# Changes in the Gut Microbiota are Associated with Hypertension, Hyperlipidemia, and Type 2 Diabetes Mellitus in Japanese Subjects

**DOI:** 10.3390/nu12102996

**Published:** 2020-09-30

**Authors:** Tomohisa Takagi, Yuji Naito, Saori Kashiwagi, Kazuhiko Uchiyama, Katsura Mizushima, Kazuhiro Kamada, Takeshi Ishikawa, Ryo Inoue, Kayo Okuda, Yoshimasa Tsujimoto, Hiromu Ohnogi, Yoshito Itoh

**Affiliations:** 1Molecular Gastroenterology and Hepatology, Graduate School of Medical Science, Kyoto Prefectural University of Medicine, Kyoto 602-8566, Japan; ynaito@koto.kpu-m.ac.jp (Y.N.); skashiwa@koto.kpu-m.ac.jp (S.K.); k-uchi@koto.kpu-m.ac.jp (K.U.); mizusima@koto.kpu-m.ac.jp (K.M.); k-kamada@koto.kpu-m.ac.jp (K.K.); iskw-t@koto.kpu-m.ac.jp (T.I.); yitoh@koto.kpu-m.ac.jp (Y.I.); 2Department for Medical Innovation and Translational Medical Science, Graduate School of Medical Science, Kyoto Prefectural University of Medicine, Kyoto 602-8566, Japan; 3Laboratory of Animal Science, Department of Applied Biological Sciences, Faculty of Agriculture, Setsunan University, Osaka 573-0101, Japan; ryo.inoue@setsunan.ac.jp; 4Contract Development & Manufacturing Center 4, Takara Bio Inc., Shiga 525-0058, Japan; okudakz@takara-bio.co.jp (K.O.); tsujimotoy@takara-bio.co.jp (Y.T.); onogih@takara-bio.co.jp (H.O.)

**Keywords:** gut microbiota, hypertension, hyperlipidemia, type 2 diabetes mellitus

## Abstract

The human gut microbiota is involved in host health and disease development. Therefore, lifestyle-related diseases such as hypertension (HT), hyperlipidemia (HL), and type 2 diabetes mellitus (T2D) may alter the composition of gut microbiota. Here, we investigated gut microbiota changes related to these diseases and their coexistence. This study involved 239 Japanese subjects, including healthy controls (HC). The fecal microbiota was analyzed through the isolation of bacterial genomic DNA obtained from fecal samples. Although there were no significant differences in the microbial structure between groups, there was a significant difference in the α-diversity between HC and the patients in whom two diseases coexisted. Moreover, Actinobacteria levels were significantly increased, whereas Bacteroidetes levels were significantly decreased in all disease groups. At the genus level, *Bifidobacterium* levels were significantly increased in the HL and T2D groups, as were those of *Collinsella* in all disease groups. In contrast, *Alistipes* levels were significantly lower in the HL group. Furthermore, metabolic enzyme families were significantly increased in all disease groups. Interestingly, the structure and function of the gut microbiota showed similar profiles in all the studied diseases. In conclusion, several changes in the structure of the gut microbiota are associated with T2D, HT, and HL in Japanese subjects.

## 1. Introduction

Hypertension (HT), hyperlipidemia (HL), and type 2 diabetes mellitus (T2D) have become major public health issues throughout the world, and are important risk factors for cerebral and cardiovascular diseases in Japan, as well as in Western countries [1,2,3,4]. It has been reported that the prevalence of HT, HL, and T2D in Japan is 15.2%, 6.9%, and 4.8% [5], respectively. In addition, it is well known that these diseases are prone to happen as comorbid conditions.

Growing clinical and animal research-based data surrounding the human gut microbiome have revealed the importance of the gut microbiome in homeostatic functions throughout the host body, and also that its composition is linked to host health and disease development [6]. Several studies have demonstrated that an altered composition of commensal bacteria is associated with the development of inflammatory bowel diseases and irritable bowel syndrome, as well as allergies and obesity in humans. Furthermore, gut microbiota dysbiosis has been recognized as having a key importance in lifestyle-related diseases.

Several studies have reported that individuals with HT, HL, and T2D, as well as those with comorbid conditions, have alterations in their intestinal microbiota profiles [7]. Qin et al. demonstrated that Chinese patients with T2D were characterized by a moderate degree of gut microbial dysbiosis, a decrease in the abundance of some universal butyrate-producing bacteria, and an increase in various opportunistic pathogens [8]. Moreover, the risk of developing T2D and HL has been reported to be correlated with an alteration in the composition and the function of the intestinal microbiota and the alleviation of hyperglycemia and hypercholesteremia by drug treatment, which shifts the gut microbiota structure, favoring beneficial bacteria such as *Blautia* and *Faecalibacterium* [9]. Regarding the gut microbiota in patients with HT, opportunistic pathogenic taxa such as *Klebsiella*, *Streptococcus*, and *Parabacteroides merdae* were widely distributed, whereas short-chain fatty acid producers such as *Roseburia* and *Faecalibacterium prausnitzii* were less abundant [10].

Nishijima et al. clearly demonstrated that the gut microbiome of the Japanese population is considerably different from that of other populations, and some studies have shown that compositional microbial changes in diseased Japanese subjects are linked to T2D [11]. Hashimoto et al. reported that the gut microbiota and its functional profiles in Japanese patients with T2D were significantly different from those in healthy individuals, and that sucrose intake was closely associated with these differences [12]. However, the detailed characteristics of gut microbiota in Japanese patients with HT, HL, T2D, and those with comorbid conditions remain unclear. In the present study, we investigated the profiles of gut microbiota in Japanese patients with these three diseases and found that several changes in the structure of the gut microbiota are associated with them.

## 2. Materials and Methods

### 2.1. Study Population

From November 2016 to April 2017, we prospectively selected 239 subjects from our outpatient clinic. Eligible subjects included male and female individuals aged between 20 and 90 years. HT was defined as systolic blood pressure ≥140 mmHg, diastolic blood pressure ≥90 mmHg, or current use of antihypertensive drugs. HL was defined as serum low-density lipoprotein cholesterol concentration ≥140 mg/dL, high-density lipoprotein cholesterol concentration <40 mg/dL, triglyceride concentration ≥150 mg/dL, or current use of agents against HL. T2D was defined as fasting plasma glucose level ≥126 mg/dL, hemoglobin A1c level ≥6.5%, or current use of antidiabetic drugs. Healthy controls (HC) did not have any gastrointestinal inflammatory diseases such as inflammatory bowel disease or functional gastrointestinal disorders such as functional dyspepsia and irritable bowel syndrome. Additional HC exclusion criteria included medication of antibiotics, corticosteroids, immunosuppressants, or acid-suppressing agents (proton pump inhibitors and histamine-type 2 receptor blockers) within the past three months, as well as a history of underlying malignant disease. In addition, patients with serious metabolic, respiratory, cardiologic, renal, hepatic, hematologic, neurologic, or psychiatric dysfunction and who regularly used medications that affect intestinal motility such as laxatives, antidepressants, opioid narcotic analgesics, anticholinergics, prokinetic agents, and prebiotics or probiotics were excluded. Subjects who were pregnant or lactating were also excluded. About a third of the subjects enrolled in this study were also involved with a previous study regarding the analysis of gut microbiota in healthy subjects and patients with T2D [12].

Overall, 54 HC, 97 patients with HT, 96 patients with HL, and 162 patients with T2D were enrolled in this study (Table 1). As shown in Figure 1, 72 patients had two of these diseases, namely, 36 patients had T2D and HL, 31 patients had T2D and HT, and 5 patients had HL and HT, and were classified as the RISK2 group. In addition, 49 patients had all three diseases and were classified as the RISK3 group, and 64 patients had only one disease (46 patients with T2D, 6 patients with HL, and 12 patients with HT) and were classified as the RISK1 group.

### 2.2. Sample Collection and DNA Extraction

Fecal samples were collected and gut bacterial composition analysis was performed according to previous reports [13,14,15]. Fecal samples the size of a grain of rice were collected using guanidine thiocyanate solution (Feces Collection kit; Techno Suruga Lab, Shizuoka, Japan). After vigorous mixing, the samples were stored at a temperature no higher than room temperature for a maximum of seven days until DNA extraction took place.

Genomic DNA was isolated using the NucleoSpin Microbial DNA Kit (Macherey-Nagel, Düren, Germany). Approximately 500 µL of each stored fecal sample was placed in a microcentrifuge tube containing 100 µL of Elution Buffer (EB). The mixture was then placed into a NucleoSpin Beads Tube with proteinase K and subjected to disruption with mechanical beads for 12 min at 30 Hz in the TissueLyser LT (Qiagen, Hilden, Germany). The subsequent extraction procedure was performed per the manufacturer’s instructions. Extracted DNA samples were purified using the Agencourt AMPure XP (Beckman Coulter, Brea, CA, USA).

### 2.3. Sequencing of the 16S rRNA Gene

Two-step polymerase chain reactions (PCRs) were performed in the purified DNA samples to obtain sequence libraries. The first PCR was performed for amplification and used a 16S (V3–V4) Metagenomic Library Construction Kit for NGS (Takara Bio Inc., Kusatsu, Japan) with primer pairs 341F (5′-TCGTCGGCAGCGTCAGATGTGTATAAGAGACAGCCTACGGGNGGCWGCAG-3′) and 806R (5′-GTCTCGTGGGCTCGGAGATGTGTATAAGAGACAGGGACTACHVGGGTWTCTAAT-3′), corresponding to the V3–V4 region of the 16S rRNA gene. The second PCR was performed to add the index sequences for the Illumina sequencer with a barcode sequence using the Nextera XT Index kit (Illumina, San Diego, CA, USA). The prepared libraries were subjected to the sequencing of 250 paired-end bases using the MiSeq Reagent v3 kit and the MiSeq (Illumina) at the Biomedical Center at Takara Bio.

### 2.4. Microbiome Analysis

The processing of sequence data, including assembly, chimera check, operational taxonomic unit (OTU) definition, and taxonomy assignment, was performed using CD-HIT-OTU version 0.0.1 and QIIME version 1.8.0 [16,17]. The taxonomy assignment of the resulting OTU was completed using RDP classifier version 2.2 and the Greengenes database [18,19]. Statistical differences (*p* < 0.05) in the relative abundance of bacterial phyla and genera among groups were evaluated using Wilcoxon rank-sum tests and the Benjamini–Hochberg method. These data have been deposited with links to BioProject accession number PRJDB10610 in the DDBJ BioProject database.

The α-diversity indices (observed species (OTU richness estimation) and Shannon index (OTU evenness estimation)) were calculated by QIIME version 1.8.0 and were statistically analyzed using Student’s unpaired *t*-tests. The *β*-diversity was estimated using the UniFrac metric to calculate the distances between the samples, and was visualized by principal coordinate analysis (PCoA).

Potential changes in the microbiome at the functional level were evaluated from sequence data downsampled to 10,000 reads using PICRUSt version 1.0.0, which uses 16S-rRNA sequence profiles to estimate metagenome content based on reference bacterial genomes and the Kyoto Encyclopedia of Genes and Genomes (KEGG) database [20,21]. The results were further statistically analyzed by Wilcoxon rank-sum tests and the Benjamini–Hochberg method. *p*-values (<0.05) were used to determine statistically significant differences between the groups.

Statistical analysis was performed using R (version 3.4.3) and the final figures were generated with R package ggplot2 or Excel.

### 2.5. Ethical Statements

This study conformed to the code of ethics stated in the Declaration of Helsinki. The Ethics Committee of the Kyoto Prefectural University of Medicine approved the research protocol (permission no. ERB-C-534-6) and all participants provided written informed consent before enrollment. The study was registered at the University Hospital Medical Information Network Center (UMIN000019486).

## 3. Results

### 3.1. Microbiota Diversity in Patients with HT, HL, and T2D

Initially, we evaluated the diversity of gut microbiota using different α-diversity indices (the observed species (OTU richness estimation) and the Shannon index (OTU evenness estimation)). They showed no statistically significant differences between HC and the patients with each disease (Figure 2A,B), though both indices showed statistically significant differences between HC and the RISK2 group. Subsequently, the overall structure of the gut microbiome for HC and the patients with these diseases using *β*-diversity indices was calculated for unweighted and weighted UniFrac distances (Figure 3). PCoA revealed that there were no microbial structural differences between HC and the patients with these diseases in unweighted and weighted distances using the two principal components (PC1 and PC2).

### 3.2. Microbiota Structure in Patients with T2D, HL, and HT

The differences in the gut microbial structure in each group were taxonomically evaluated at the phylum level (Figure 4). In agreement with previous results, the microbiota composition included four predominant phyla: Firmicutes, Bacteroidetes, Actinobacteria, and Proteobacteria. Interestingly, the Actinobacteria phylum levels showed a statistically significant increase in all patient groups compared to HC. On the other hand, the Bacteroidetes phylum showed significantly decreased levels in all patient groups except in the RISK2 group compared to HC, although the abundance of Bacteroidetes phylum tended to decrease in the RISK2 group, as well as in other groups.

Taxonomic changes in the microbial community were also evaluated at the genus level. As presented in Figure 5, the abundance of four genera showed significant differences between HC and the patients with the diseases. The levels of the *Bifidobacterium* genus were significantly increased in the HL, T2D, RISK1, and RISK2 groups compared to HC, and those of the *Collinsella* genus were also significantly increased in the HT, HL, T2D, RISK2, and RISK3 groups compared to HC, whereas the *Escherichia* genus had its levels significantly increased only in the RISK3 group. In contrast, the levels of the *Alistipes* genus were significantly decreased in the HL group compared to HC. Interestingly, these alterations tended to be similar in all groups.

Finally, we evaluated potential differences in the function of the microbiome using the PICRUSt software (Figure 6). Microbiome functions showed a similar pattern among patients with HT, HL, and T2D. When comparing these disease groups with HC at the second level of KEGG, the proportion of genes responsible for membrane transport and metabolic enzyme families was significantly increased in the HL and T2D groups, and in the HT, HL, and T2D groups, respectively, compared to HC.

## 4. Discussion

In the present study, we analyzed the fecal gut microbiota in Japanese subjects with HT, HL, and T2D. Interestingly, these three diseases showed similar profiles of gut microbiota structure and functions. In addition, as lifestyle-related diseases, they often overlap, and such cases also showed similar gut microbiota profiles in our study. Importantly, the results of our functional analysis also presented a similar trend between HT, HL, and T2D.

The levels of the Actinobacteria phylum were significantly increased in patients with HT, HL, and T2D, and this increase was reflected in the increased abundance of the *Bifidobacterium* genus. Similarly, for comorbid conditions, the levels of the Actinobacteria phylum and *Bifidobacterium* genus were also significantly increased. Conversely, the abundance of the Bacteroidetes phylum was significantly decreased in patients with HT, HL, T2D, and comorbid conditions. In agreement with our results, Adachi et al. demonstrated that T2D patients had larger and smaller fecal populations of *Bifidobacterium* and *Bacteroides*, respectively, than the control individuals [22]. Interestingly, they also demonstrated that T2D patients had consumed more carbohydrates and had lower fecal propionate and butyrate concentrations, and that the levels of *Bifidobacterium* were negatively correlated with the carbohydrate intake. Hashimoto et al. also found that the Actinobacteria phylum was highly abundant in patients with T2D, whereas the Bacteroidetes phylum was less abundant [12]. In addition, the levels of diabetic-type gut microbes were altered by sucrose intake at the genus level, specifically *Bacteroides* and *Bifidobacterium* [12]. Therefore, a rich abundance of the Actinobacteria phylum and a lower abundance of the Bacteroidetes phylum seem to be characteristic of the gut microbiota of Japanese patients with T2D.

In the present study, the abundance of the *Collinsella* genus was also significantly increased in the HT, HL, T2D, RISK2, and RISK3 groups compared to HC. It has been reported that the *Collinsella* genus, which belongs to the Actinobacteria phylum, is associated with serum cholesterol and triglyceride levels [23]. Although the precise molecular mechanisms by which *Collinsella* affects host metabolism are unknown, its high abundance has been associated with obesity and T2D [24,25]. Interestingly, low dietary fiber might enable the overgrowth of *Collinsella,* and a structured weight loss program could significantly decrease *Collinsella* levels in patients with obesity and T2D [26,27].

In contrast, the abundance of the *Alistipes* genus was significantly decreased in the HL group compared to HC. The abundance of *Alistipes*, which belongs to the Bacteroidetes phylum, was lower in rodent models with high fat diet-induced hyperlipidemia [28,29]. In addition, the levels of this genus were also significantly decreased in obese Mexican individuals [30]. Meanwhile, they were positively correlated with systolic and diastolic blood pressure in Chinese hypertensive patients with treatment-naive hypertension [31].

This study has several limitations. The clinical investigation was performed with a limited number of subjects in a single center, and the analysis was performed on both patients being treated for HT, HL, and T2D, and treatment-naive patients. This limitation may have affected the results because patients with these diseases often improve their dietary behaviors and control their intake of salt, sugar, and oil, and some therapeutic agents for these conditions, such as metformin, affect the structure of gut microbiota [5]. In addition, as we did not obtain the daily data of dietary intakes in this investigation, we could not analyze the association between the gut microbiota and dietary intakes. A detailed evaluation of the gut microbiota in a larger number of patients with these lifestyle-related diseases is needed in the near future.

## 5. Conclusions

In conclusion, this study demonstrated that several compositional changes in the gut microbiota are associated with lifestyle-related diseases such as HT, HL, and T2D, and that the gut environment seems to share a common pattern in these three diseases. Our data also highlight the crucial importance of gut microbiota through a better understanding of intestinal functions, and that patients with HT, HL, and T2D have gut dysbiosis that may consequently contribute to disease onset and influence its clinical prognosis. Furthermore, homeostatic disturbances in the gut-related metabolism may underlie the pathogenesis of these diseases.

## Figures and Tables

**Figure 1 nutrients-12-02996-f001:**
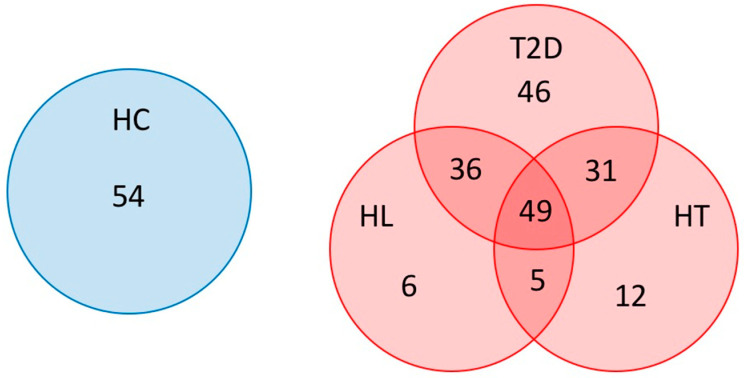
Subjects enrolled in this study. Patients overlapping T2D, HT, and HL were indicated using a Venn diagram. (HC: healthy controls, T2D: type 2 diabetes, HT: hypertension, HL: hyperlipidemia).

**Figure 2 nutrients-12-02996-f002:**
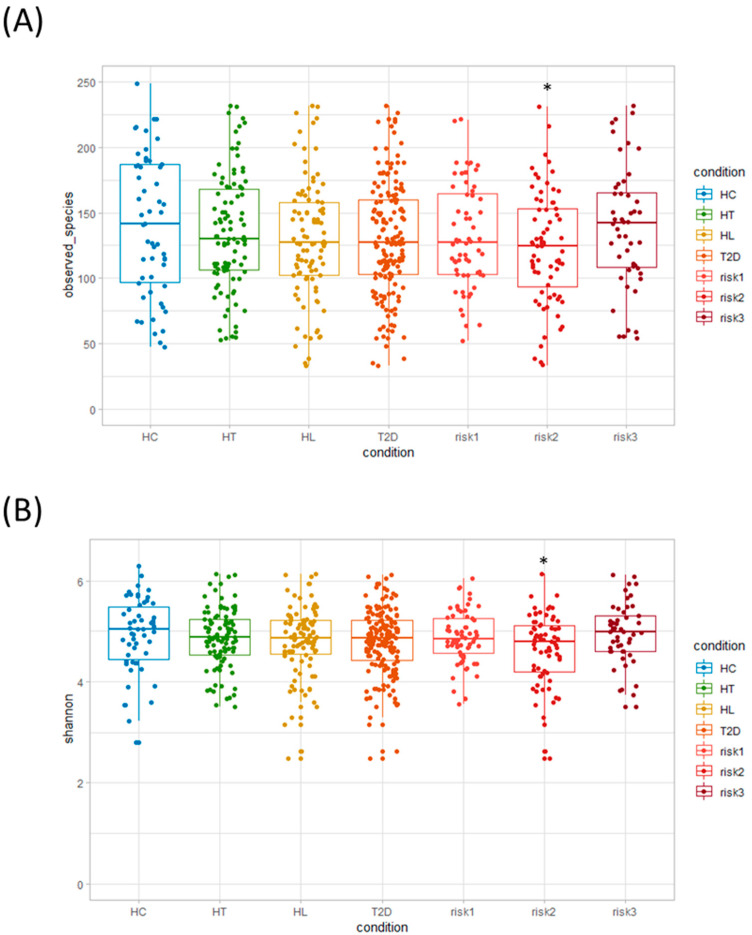
Analysis of α-diversity in gut microbiota. We evaluated and compared α-diversity indices: (**A**) observed species (operational taxonomic unit (OTU) richness estimation) and (**B**) Shannon index (OTU evenness estimation), using Student’s unpaired *t*-tests (* *p* < 0.05 vs. HC). (HC: healthy controls, HT: hypertension, HL: hyperlipidemia, T2D: type 2 diabetes, RISK1: patients with only one disease among HT, HL, and T2D, RISK2: patients with two diseases among HT, HL, and T2D, RISK3: patients with the three diseases (HT, HL, and T2D)).

**Figure 3 nutrients-12-02996-f003:**
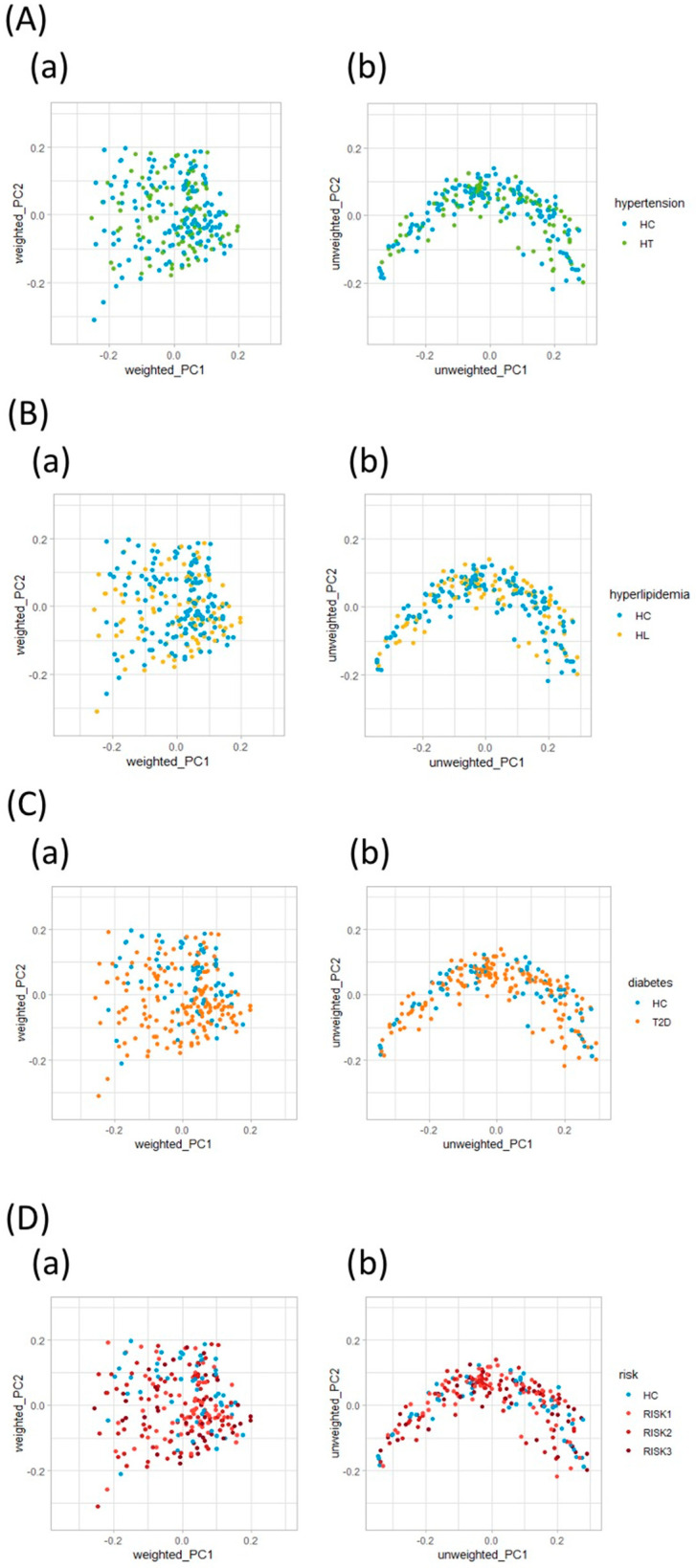
Principal coordinate analysis plots of gut microbiota. Distances were calculated with unweighted (**a**) and weighted (**b**) UniFrac for HT (**A**), HL (**B**), T2D (**C**), and RISK1, 2, and 3 (**D**) compared to HC. (HC: healthy controls, HT: hypertension, HL: hyperlipidemia, T2D: type 2 diabetes, RISK1: patients with only one disease among HT, HL, and T2D, RISK2: patients with two diseases among HT, HL, and T2D, RISK3: patients with the three diseases (HT, HL, and T2D)).

**Figure 4 nutrients-12-02996-f004:**
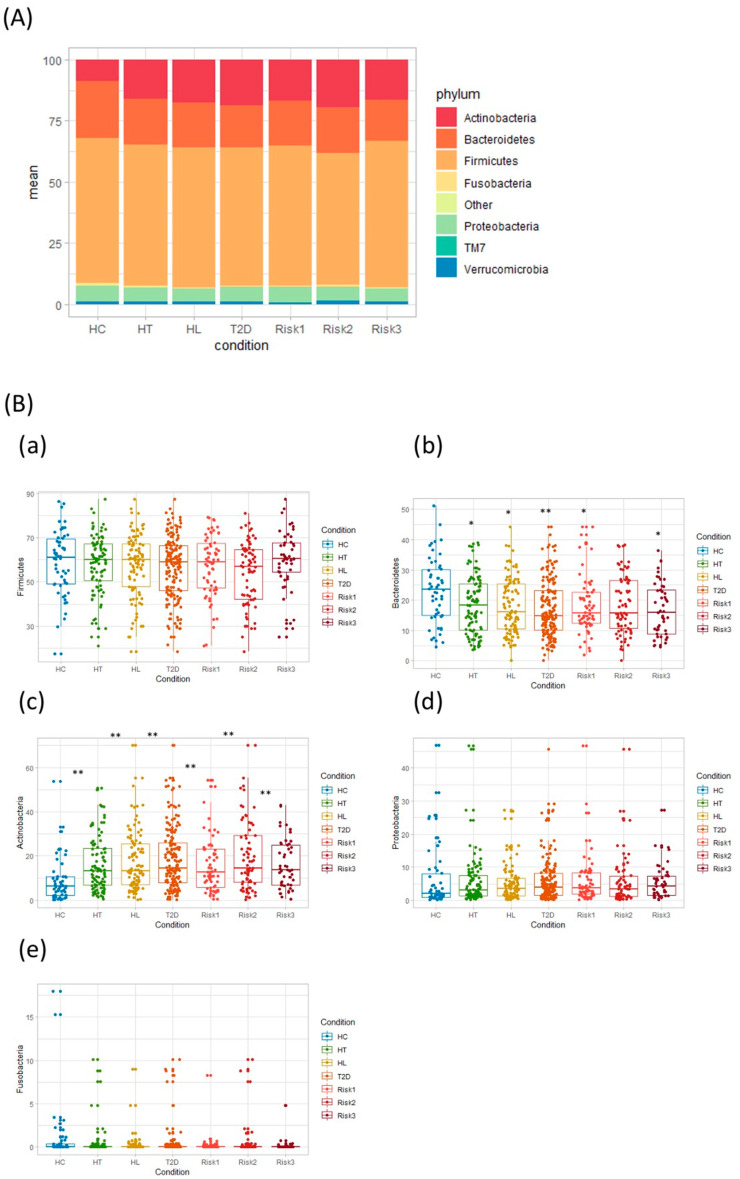
Comparative analysis of the taxonomic composition of the microbial community at the phylum level. Each component of the cumulative bar chart indicates a phylum (**A**). Each participant group was evaluated for Firmicutes (**B**-**a**), Bacteroidetes (**B**-**b**), Actinobacteria (**B**-**c**), Proteobacteria (**B**-**d**), and Fusobacteria (**B**-**e**) using Student’s unpaired *t*-tests (* *p* < 0.05, ** *p* < 0.01 vs. HC). (HC: healthy controls, HT: hypertension, HL: hyperlipidemia, T2D: type 2 diabetes, RISK1: patients with only one disease among HT, HL, and T2D, RISK2: patients with two diseases among HT, HL, and T2D, RISK3: patients with the three diseases (HT, HL, and T2D)).

**Figure 5 nutrients-12-02996-f005:**
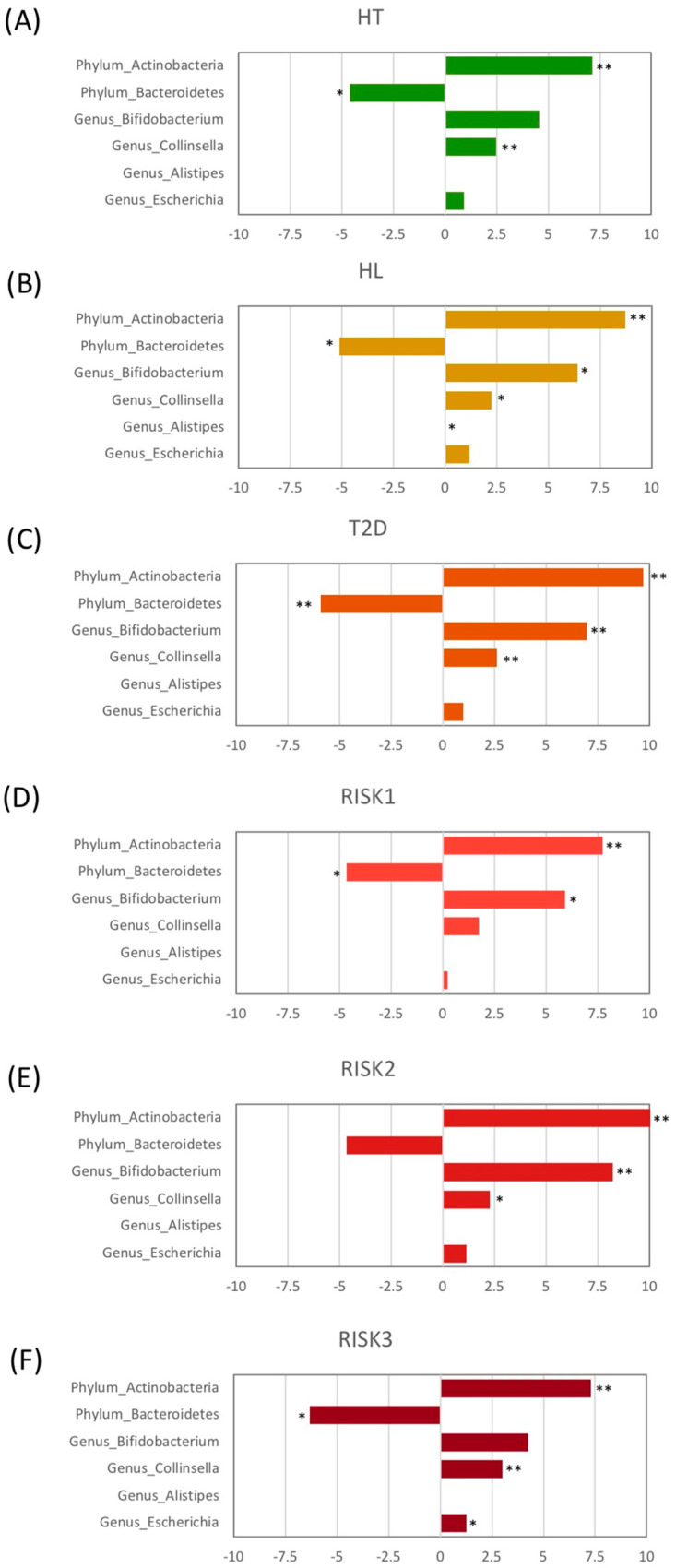
Relative abundance of gut microbiota at the genus level. The genera with significant differences among each group compared to HC are presented for HT (**A**), HL (**B**), T2D (**C**), RISK1 (**D**), RISK2 (**E**), and RISK3 (**F**), using Student’s unpaired *t*-tests (* *p* < 0.05, ** *p* < 0.01 vs. HC). (HC: healthy controls, HT: hypertension, HL: hyperlipidemia, T2D: type 2 diabetes, RISK1: patients with only one disease among HT, HL, and T2D, RISK2: patients with two diseases among HT, HL, and T2D, RISK3: patients with the three diseases (HT, HL, and T2D)).

**Figure 6 nutrients-12-02996-f006:**
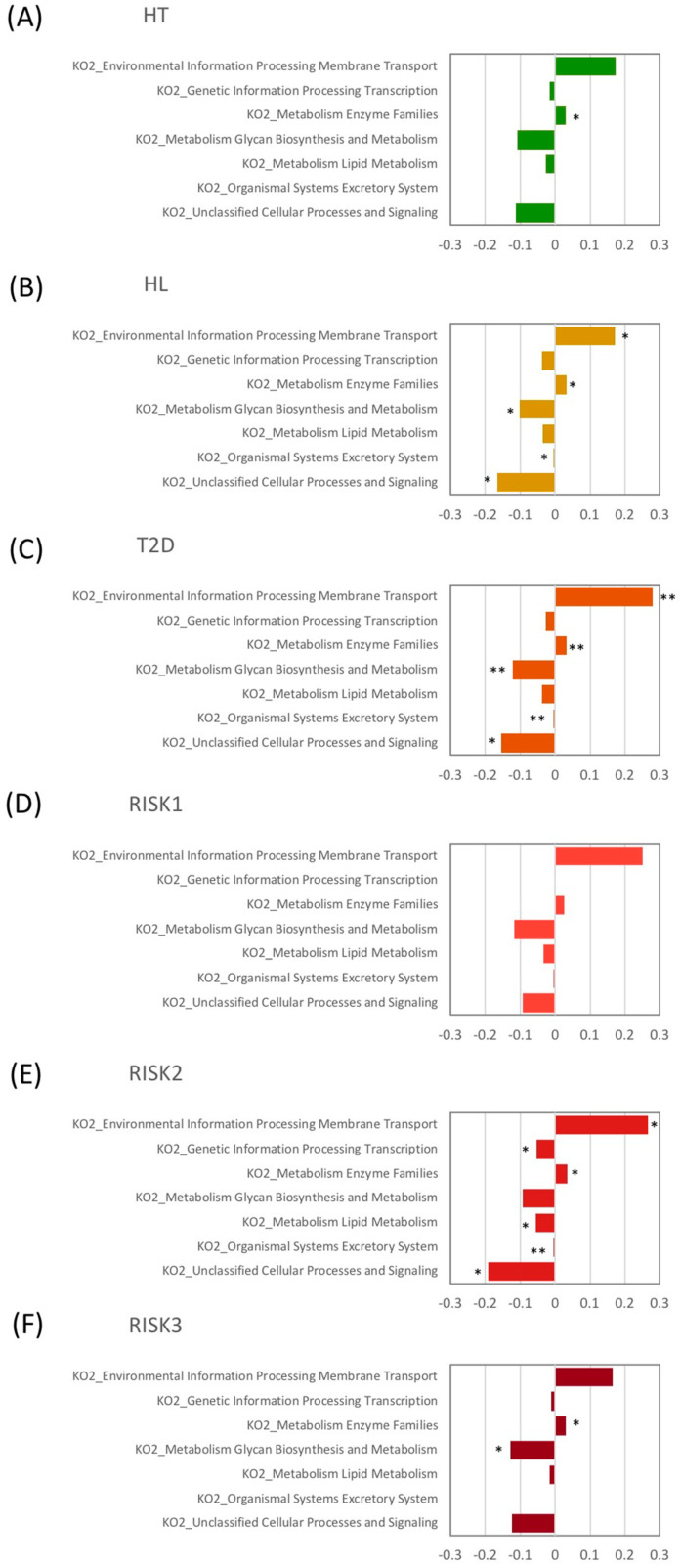
Relative abundance of functional pathways. The KEGG (Kyoto Encyclopedia of Genes and Genomes) database functional categories at the second level of KEGG and metabolic enzyme families are shown in the histograms for HT (**A**), HL (**B**), T2D (**C**), RISK1 (**D**), RISK2 (**E**), and RISK3 (**F**) compared to HC, using Student’s unpaired *t*-tests (* *p* < 0.05, ** *p* < 0.01 vs. HC). (HC: healthy controls, HT: hypertension, HL: hyperlipidemia, T2D: type 2 diabetes, RISK1: patients with only one disease among HT, HL, and T2D, RISK2: patients with two diseases among HT, HL, and T2D, RISK3: patients with the three diseases (HT, HL, and T2D)).

**Table 1 nutrients-12-02996-t001:** Baseline characteristics of enrolled subjects in this study.

	ALL	Healthy Control	Hypertension	Hyperlipidemia	Type 2 Diabetes
Number of samples	239	54	97	96	162
Male/Female	113/126	21/33	49/48	51/45	79/83
Age (median)	68 (16–88)	65.5 (16–88)	69 (37–87)	69 (37–87)	69 (37–87)
Height (median)	158.9 (135.0–185.3)	159.5 (140.0–178.0)	160.0 (135.0–180.5)	160.7 (140.0–180.5)	160.0 (140.0–185.3)
Weight (median)	54.80 (33.10–128.60)	53.75 (33.10–85.90)	61.50 (35.65–128.60)	62.60 (39.0–128.60)	61.40 (39–128.60)

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
