# Peer review of "Changes in the Gut Microbiota are Associated with Hypertension, Hyperlipidemia, and Type 2 Diabetes Mellitus in Japanese Subjects"

_nutrients, 2020, doi:10.3390/nu12102996_

Round 1
Reviewer 1 Report
Tagaki et al report changes in diversity in Japanese with hypertension, hyperlipidemia and T2D compared with healthy controls. I have a few questions and suggestions for the authors.
p. 2, line 7 from bottom What were the “other factors”
p. 4, line 14 Insert the BioProject Accession number
How does one interpret the fact that there were no significant differences in alpha diversity (OTU richness and OTU evenness) between healthy controls and each disease group with the exception of healthy controls versus the RISK2 group? Is this explained by the differences in microbial structure at the phylum and genus levels? I found this a bit unclear.
Discussion: Reference is made to the effects of diet, serum lipids, dietary fibre, hyperlipidemia, etc on gut microbiota. However, possible links between gut microbiota diversity and for dietary intakes, blood lipids, body weight, age etc. are difficult to interpret because data on these factors is absent. Do you have some of these data for your participants and if so, do these factors relate to differences in diversity or other descriptive measure of gut microbiota? Other questions come to mind when trying to understand the observed differences. For example, were there differences in dietary intake among the groups, did this change in an age-related manner? Were some participants on weight loss or exercise programs? Were there differences in drug intake? Presumably, subjects in some groups would be taking drugs to treat their condition(s) such as metformin in T2D.
Overall, the paper is well written, the study was performed well and the interpretation of the results is reasonable. The authors suggest that this study provides novel data with respect to the Japanese population. I am not qualified to judge this claim.
Author Response
To reviewer 1
Thank you for your comments. As you kindly suggested, our manuscript was revised as follows.
1, p. 2, line 7 from bottom What were the “other factors”
(Response) Thank you for your constructive comments. We described this sentence to avoid the enrollment of subjects who took the agents for improving the gastrointestinal motility. However, to avoid reader’s confusion according to your comments, in addition, as the high duplication rates of this paragraph from editor, we revised the description of this paragraph as follows;
(lines 71-87) From November 2016 to April 2017, we prospectively selected 239 subjects from our outpatient clinic. Eligible subjects included male and female subjects aged between 20 and 90 years. HT was defined as systolic blood pressure ≥140 mmHg, diastolic blood pressure ≥90 mmHg, or current use of antihypertensive drugs. HL was defined as serum low-density lipoprotein cholesterol concentration ≥140 mg/dL, high-density lipoprotein cholesterol concentration <40 mg/dL, triglyceride concentration ≥150 mg/dL, or current use of agents against HL. T2D was defined as a fasting plasma glucose level ≥126 mg/dL, hemoglobin A1c level ≥6.5%, or current use of anti-diabetic drugs. Healthy controls (HC) did not have any gastrointestinal inflammatory diseases such as inflammatory bowel disease or functional gastrointestinal disorders such as functional dyspepsia and irritable bowel syndrome. Additional HC exclusion criteria were as follows: medication of antibiotics, corticosteroids, immunosuppressants, or acid-suppressing agents (proton pump inhibitors and histamine-type 2 receptor blockers) within the past three months, and a history of underlying malignant disease. In addition, patients with serious metabolic, respiratory, cardiologic, renal, hepatic, hematologic, neurologic, or psychiatric dysfunction and who regularly used medications that affect intestinal motility such as laxatives, antidepressants, opioid narcotic analgesics, anticholinergics, prokinetic agents, and prebiotics or probiotics were excluded. Subjects who were pregnant or lactating were also excluded.
2, p. 4, line 14 Insert the BioProject Accession number
(Response) Thank you for your kind comments. Now, we have registered BioProject and obtain Accession number (PRJDB10610). So, we revised thisdescription as follows;
(line 136) links to BioProject accession number PRJDB10610
3, How does one interpret the fact that there were no significant differences in alpha diversity (OTU richness and OTU evenness) between healthy controls and each disease group with the exception of healthy controls versus the RISK2 group? Is this explained by the differences in microbial structure at the phylum and genus levels? I found this a bit unclear.
(Response) Thank you for your important comments. As your comments, we also surprised at no significant differences in alpha diversity (OTU richness and OTU evenness) between healthy controls and each disease group with the exception of healthy controls versus the RISK2 group. The other group except the RISK2 group seems to have a significant difference when looking only at the average value, but there was no significant difference due to the large variance. In addition, we performed various analysis in the phylum, genus, and OTU level, but we had no clear results have been obtained to explain this phenomenon. We discussed with another all authors, and we had no description regarding this statistic issue to avoid the reader’s confusion.
4, Discussion: Reference is made to the effects of diet, serum lipids, dietary fibre, hyperlipidemia, etc on gut microbiota. However, possible links between gut microbiota diversity and for dietary intakes, blood lipids, body weight, age etc. are difficult to interpret because data on these factors is absent. Do you have some of these data for your participants and if so, do these factors relate to differences in diversity or other descriptive measure of gut microbiota? Other questions come to mind when trying to understand the observed differences. For example, were there differences in dietary intake among the groups, did this change in an age-related manner? Were some participants on weight loss or exercise programs? Were there differences in drug intake? Presumably, subjects in some groups would be taking drugs to treat their condition(s) such as metformin in T2D.
(Response) Thank you for your kind comments. We totally agree with your direction. However, we unfortunately obtained the detailed dietary intake data in this study. In next step, we will solve this issue in near future. According to your direction, we added the sentence to the limitation in the section of discussion as follows;
(lines 261-263) In addition, as we did not obtain the daily data of dietary intakes in this investigation, we could not analyze the association between the gut microbiota and dietary intakes.
I would be most grateful to you if you would kindly inform me of your final decision on this manuscript in due course of time. 

Yours sincerely,
Tomohisa Takagi, MD, PhD.
Associate Professor
Molecular Gastroenterology and Hepatology, Graduate School of Medical Science, Kyoto Prefectural University of Medicine, Kyoto, Japan
Kawaramachi-Hirokoji, Kamigyo-ku, Kyoto 602-8566, Japan
Tel: + 81-75-251-5508
Fax: +81-75-251-0710
E-mail: takatomo@koto.kpu-m.ac.jp
Reviewer 2 Report
This is an interesting topic and the research has been conducted well within the limits described by the authors.
i second the fact that dietary changes with altered life style and medications including hypoglycemic agents can certainly be significant confounders in the results obtained. However within the limits of the study the data is presented well.
why do the authors think that RISK2 group didnot show an appropriate decrease in bacteroides phyllum level? This has not been discussed well and I urge the authors to explain this phenomenon.
the images are exceedingly small making it difficult for the reader to fully appreciate the data representation.
Author Response
Thank you for your comments. As you kindly suggested, our manuscript was revised as follows.
This is an interesting topic and the research has been conducted well within the limits described by the authors. Second, the fact that dietary changes with altered life style and medications including hypoglycemic agents can certainly be significant confounders in the results obtained. However within the limits of the study the data is presented well.
(Response) Thank you for your constructive comments. Reviewer 1 also pointed out the association between the gut microbiota and dietary intakes. Therefore, we added the sentence regarding the dietary intake to the limitation in the section of discussion as follows;
(lines 261-263) In addition, as we did not obtain the daily data of dietary intakes in this investigation, we could not analyze the association between the gut microbiota and dietary intakes.
why do the authors think that RISK2 group did not show an appropriate decrease in bacteroides phylum level? This has not been discussed well and I urge the authors to explain this phenomenon.
(Response) Thank you for your kind comments. Surely, we could not find a significant decrease in bacteroides phylum levelin RISK2 group. However, as abundance of Bacteroidetes tended to decrease in Risk2 group as well as another groups (P=0.069049). According to your suggestion, we revised this section as follows;
(lines 185-186) although the abundance of Bacteroidetes phylum tended to decrease in Risk2 group as well as another groups.
the images are exceedingly small making it difficult for the reader to fully appreciate the data representation.
(Response) Thank you for your kind suggestion. According to your suggestion, we revised the images of Figure3, 4, 5, and 6.
I would be most grateful to you if you would kindly inform me of your final decision on this manuscript in due course of time. 

Yours sincerely,
Tomohisa Takagi, MD, PhD.
Associate Professor
Molecular Gastroenterology and Hepatology, Graduate School of Medical Science, Kyoto Prefectural University of Medicine, Kyoto, Japan
Kawaramachi-Hirokoji, Kamigyo-ku, Kyoto 602-8566, Japan
Tel: + 81-75-251-5508
Fax: +81-75-251-0710
E-mail: takatomo@koto.kpu-m.ac.jp